# Lupenone, a wonder chemical obtained from *Euphorbia segetalis* to boost affinity for the transcriptional factor escalating drought-tolerance in *Solanum Lycopersicum*: A cutting-edge computational biology approach

**Sandip Debnath**[1]*, **Taha Alqahtani**[2], **Ali Alqahtani**[2], **Hanan M. Alharbi**[3], **Shopnil Akash**[4]*

**1** Department of Genetics and Plant Breeding, Institute of Agriculture, Visva-Bharati University, Sriniketan, West Bengal, India, **2** Department of Pharmacology, College of Pharmacy, King Khalid University, Abha, Saudi Arabia, **3** Department of Pharmaceutics, College of Pharmacy, Umm Al-Qura University, Makkah, Saudi Arabia, **4** Department of Pharmacy, Faculty of Allied Health Sciences, Daffodil International University, Ashulia, Dhaka, Bangladesh

* sandip.debnath@visva-bharati.ac.in (SD); shopnil.ph@gmail.com (SA)

**Data Availability Statement:** All relevant data are within the paper.

## Abstract

Drought is the single greatest abiotic factor influencing crop yield worldwide. Plants remain in one area for extended periods, making them vulnerable to natural and man-made influences. Understanding plant drought responses will help us develop strategies for breeding drought-resistant crops. Large proteome analysis revealed that leaf and root tissue proteins respond to drought differently depending on the plant's genotype. Commonly known as tomatoes, *Solanum Lycopersicum* is a globally important vegetable crop. However, drought stress is one of the most significant obstacles to tomato production, making the development of cultivars adapted to dry conditions an essential goal of agricultural biotechnology. Breeders have put quite a lot of time and effort into the tomato to increase its productivity, adaptability, and resistance to biotic and abiotic challenges. However, conventional tomato breeding has only improved drought resistance due to the complexity of drought traits. The resilience of tomatoes under drought stress has been the subject of extensive study. Using contemporary sequencing approaches like genomics, transcriptomics, proteomics, and metabolomics has dramatically aided in discovering drought-responsive genes. One of the most prominent families of plant transcription factors, WRKY genes, plays a crucial role in plant growth and development in response to natural and abiotic stimuli. To develop plants that can withstand both biotic and abiotic stress, understanding the relationships between WRKY-proteins (transcription factors) and other proteins and ligands in plant cells is essential. This is despite the fact that tomatoes have a long history of domestication. This research aims to utilize Lupenone, a hormone produced in plant roots in response to stress, to increase drought resistance in plants. Lupenone exhibits a strong affinity for the WRKY protein at -9.64 kcal/mol. Molecular docking and modeling studies show that these polyphenols have a significant role in making *Solanum Lycopersicum* drought-resistant and improving

**Funding:** The authors are grateful to the Deanship of Scientific Research at King Khalid University for funding this study through the Large Research Group Project, under grant number RGP 2/100/43.

**Competing interests:** The authors have declared that no competing interests exist.

the quality of its fruit. As a result of climate change, droughts are occurring more frequently and persisting for more extended periods, making it necessary to breed crops resistant to drought. While considerable variability for tolerance exists in wild cousins, little is known about the processes and essential genes influencing drought tolerance in cultivated tomato species.

## Introduction

Crop production faces considerable challenges from stress, salt, and drought—two aspects of climate change that are particularly damaging to sustainable agriculture. Agriculture productivity is likely to be constrained due to the atmosphere, which is expected to become significantly less favorable as a result of ongoing greenhouse gas emissions [1]. As a result, evaporative and transportive losses, as well as soil dryness and salinity, would all increase, as would the danger of insect and disease infestation. The unfavorable conditions are harmful to the plants' development, survival, and crop quality. Drought conditions are widely known to be particularly damaging to the growth of potatoes as a crop. The fundamental reason is that tomatoes have more fragile roots than other plants. When a plant is deprived of its essential amount of water, it will experience stress, which may result in the loss of leaf tissue. This mortality usually occurs in bands that radiate outward from the leaf's borders. It is probable that some individuals will confuse this with the late blight. Plants are especially prone to the detrimental effects of stress when rapidly developing or when they are subjected to other conditions that cause them to lose more water through transpiration. Under these conditions, it may be difficult to create a plan that takes advantage of the times when the ground is dry and moist. When there is a lot of humidity in the air, the rate at which water is lost through perspiration slows down. When huge plants are housed in pots that are too small for them, it might be difficult to maintain an optimum moisture level in the soil. The absence of light green, wilted border tissue around leaf symptoms, the absence of pathogen growth, and the absence of stem lesions distinguish physiological harm from late blight. In contrast, late blight can be identified by the presence of all three features. Low or high temperatures, protracted droughts or floods, excessive amounts of salt in the soil, and other abiotic conditions are some environmental variables that typically result in low yields [2]. The tomato is typically regarded as one of the most economically significant horticultural crops produced worldwide. Furthermore, it is an important crop for agricultural output in dry and semi-arid environments. Abiotic stressors, particularly those caused by a water shortage, negatively impact tomato production, which is especially true in the Mediterranean region. The processes that govern responses to water stress in this horticultural species are poorly understood, and only a few genes involved in tomato drought tolerance have been found. Even though tomatoes are incredibly significant economically, this is the case. The tomato, scientifically known as *Solanum Lycopersicum*, is a common vegetable that could be harmed if the conditions were unfavorable [3]. Tomatoes grown in open fields are more susceptible to non-living factor damage because they are closer to these factors than greenhouse-grown tomatoes. These factors include the impact of drought and extreme heat on crop production, the length of the growing season, and the amount of land available for crop cultivation. When plant roots are exposed to temperatures below freezing, they lose the majority of their ability to transport water throughout the plant. As a result, the water becomes more viscous, and the essential membrane found in the roots becomes less conductive [4]. As a result, plants can absorb less water, potentially leading to water stress in

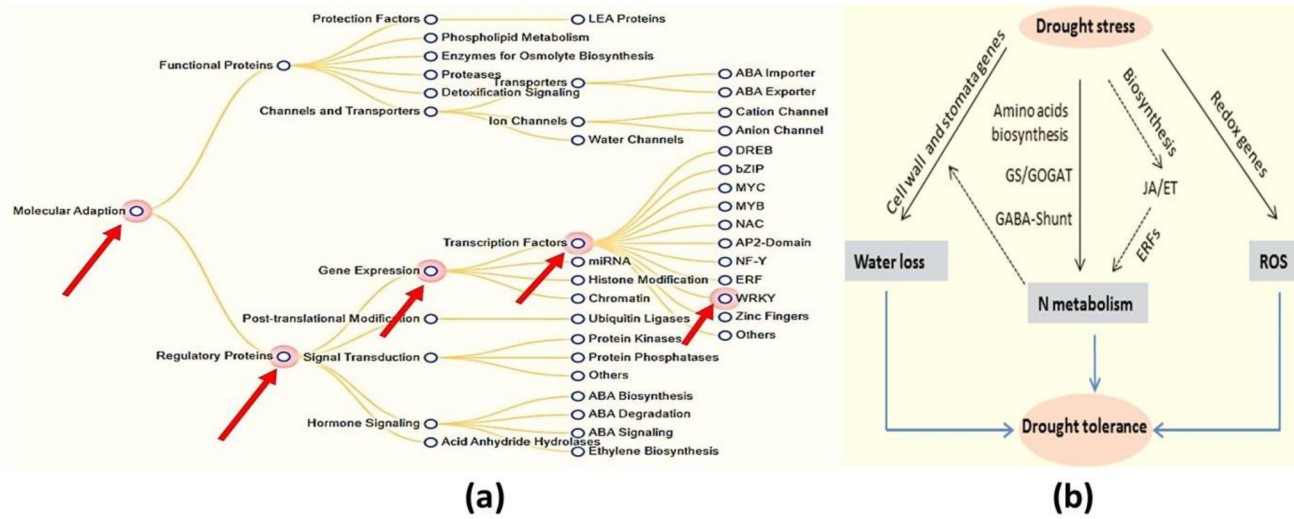

**Fig 1.** (a) Boosted lenience to drought & biotic pressure and genes in charge (*Available from*: https://pgsb.helmholtz-muenchen.de/droughtdb/drought_db.html); (b) A model for genes that regulate decreases water loss by upregulating cell wall and stomatal regulator genes during drought [17].

the shoots. As a result, the tomato leaves begin to wilt, and both photosynthesis and water loss are halted [5]. There are 5936 members of the WRKY transcription factor family discovered in various plant systems (PlantTFDB 3.0). WRKY is the sweet potato DNA-binding protein SPF1, which may help control gene expression, as described for the first time [5]. The moniker "WRKY" refers to their WRKY domain, which begins at the N-terminus with a relatively consistent sequence of WRKYGQK and ends with a Cx4-5Cx22-23HxH or Cx7Cx23HxC zinc-finger motif [4, 6]. Much research has been done on the WRKY domain, specifically how it regulates genes and communicates with other molecules within plant cells. WRKY transcription factors were previously thought to be involved in plant pathogen defense [6, 7]. However, later research has shown that they are linked to abiotic stressors [7], seed dormancy and germination [8–10], and seed development [10, 11]. The ability of most WRKY proteins to identify TTGACC/T W-box sequences in the promoter region allows them to be identified [6, 12, 13], but this does not suggest that they all perform the same function. Controlling the specificity of WRKY transcription factors thus necessitates more than simply identifying the necessary W box promoter components. This could be due to interactions with proteins involved in signal transduction, transcription, and chromatin remodeling, making transgenic tobacco more vulnerable to cold, Lupenone (L.N.), salt, and dehydration stressors [14]. There is evidence that WRKY proteins can communicate with calmodulin, a $Ca^{2+}$-binding signaling molecule. Recent proteomics research revealed that the WRKY and 14-3-3 proteins collaborate (illustrated in **Fig 1A**) [14].

Besides, drought causes a significant drop in agricultural production by disrupting growth, nitrogen and water interactions, photosynthesis, and assimilate partitioning. Plants react differently to drought stress based on species, growth stage, and other variables. High-temperature stress causes a cascade of morphological, biochemical, and physiological changes in plants, significantly affecting plant growth and development. Heat shocks caused by increasing air temperatures are now a critical limiting factor to agricultural output worldwide. Changes in the timing and location of harvests might result from this temperature increase. Damage to proteins, disruption of protein synthesis, inactivation of critical enzymes, and membrane damage are all possible outcomes of prolonged exposure to high temperatures. Heat stress may have profound impacts on cell division. These problems harm plants because they may stunt

their development and promote more oxidative damage. In addition, short exposure to high temperatures during seed filling may hasten filling, leading to poor grade and a lower yield [15, 16].

During the rapid development phase of drug discovery, natural compounds based on the low toxicity profile of small-molecule inhibitors could be beneficial. Squalene, created via farnesol pyrophosphate tail-tail condensation, is cyclized to create terpenoids. There are numerous biological uses for terpenoids. There are innumerable triterpenoids in plants. The majority of triterpenes include 30 carbon atoms and six isoprene units. Triterpenoids and saponins are present in both single-celled and double-celled plants. *Compositae*, *Leguminosae*, *Euphorbiaceae*, *Meliaceae*, *Euonymus*, *Rubiaceae*, Olive Branch, and Labiatae are among the plant families that contain free triterpenes [3]. More than 30 species have skeletons made of triterpenoid molecules. Four or five rings make up the majority of triterpenes. The majority of the pharmacological and biological activities of pentacyclic triterpenes involve lowering inflammation, safeguarding the liver, battling cancer, and altering the immune syste [18, 19]. Pentacyclic triterpenoids include -amyrin, -amyrin, oleanane, friedelin, lupane, and hopane [19]. The lupine type of triterpenoids has the most basic structure. E is a carbon ring with just five members compared to the six-membered carbon rings A, B, C, and D. The E ring's 19th position is changed to an isopropyl group for the setup. Of the three ketones, only the lupine-type triterpenoid nuclear ring possesses connected double bonds at positions 1 and 2. Having a ketone group in the third ring position, Lupenone is a polar triterpenoid of the lupane type that is rare in compounds [20]. Dichloromethane is combined with lupeol, P.C.C., and P.C.C. at room temperature to create Lupenone [9]. Doctors and researchers are interested in Lupenone because of its pharmacological properties; it's found in *Euphorbia segetalis* [21]. Lupenone may treat Chagas disease, viral infections, diabetes, and inflammation. However, the majority of research on lupenone's anti-viral and anti-tumor activities is conducted in a laboratory, and in-vivo investigations are still required to verify the results of the lab studies. This research has demonstrated the various ways Lupenone can aid in the drought struggle in *Solanum Lycopersicum*.

Molecular Docking is the method most frequently used in structure-based ligand design because it allows one to consider the action of small molecules in particular protein targets and predict the structure of the ligand-receptor complex. The goal of this study was to search the PubChem database for Lupenone, a novel chemical that could improve drought resistance by blocking the WRKY gene's transcription factor. Researchers used virtual screening and structure-based docking approaches to determine the molecule that effectively inhibits a target. The compound of interest (the target compound) and the compound that performed the best all-around was examined for structural stability in this study. Studies using molecular dynamics simulations were used in the investigation. To make *Solanum Lycopersicum* a viable contender in the battle against drought resistance, the main objective of this research is to identify a novel chemical that acts as a potent enhancer of drought resistance in *Solanum Lycopersicum*.

## Resources and methods

### Potential target preparation

Accessible from the RCSB protein data bank (PDB) (https://www.rcsb.org/2AYD) are three-dimensional tertiary structures of the β-Secretase protein for study. The structure was imported using a freely available molecular editor (Discovery studio visualizer 4.0) [22]. Co-crystal ligands and heteroatoms were deleted before the structure was saved in.pdb format. The Chimera UCSF team employed a thousand-step steepest-descent and a thousand-step Conjugate gradient of energy minimization approach for this optimization. Lupenone (**Chem**

**I. D: 92158**) was downloaded as a.sdf file from PubChem. These.pdb files result from a ligand structure loaded into the Discovery Studio visualizer.

## Virtual screening of selected compounds

The active site of an enzyme is a portion of the enzyme that possesses a unique structure that enables it to form a stable bond with a particular type of molecular substrate [23, 24]. Because of this, a chemical reaction is triggered in the enzyme. The "active site" of an enzyme is a portion of the enzyme that is shaped in a certain way that allows it to attach to a particular type of molecular substrate. AS helps chemical compounds establish enough contact sites so that they can bind strongly to the enzymes of choice. This allows AS to ensure that the microenvironments for catalysis are adequate and ideal. Therefore, to establish a solid connection between our chemical and the active side of the protein, we examined the active side of the protein using BIOVIA Discovery Studio Visualizer version 19.1.0.18287. This was done so that we could achieve our goal. This was done in order to create a strong binding affinity in the end product. The binding site in the protein complex was also discovered and utilized in creating the receptor grid by using the virtual screening application AutoDock Vina. A virtual high-throughput screening of ten distinct compounds was carried out with AutoDock Vina 4.2.6. The compounds were chosen based on their best binding energy scores with the macromolecule with **PDB ID: 2AYD**. The best-docked posture with the highest binding energies was selected for re-docking and additional research out of the nine possible poses for each ligand, ranked from top to lowest according to the binding energies they produced.

## Molecular docking studies

In AutoDock M.G.L. tool 1.5.6, the receptor protein for Docking was produced. To construct the receptor grid, residues around 2Å the 2AYD were linked to the co-crystal of Lupenone were utilized. The receptors and ligands were saved in the pdbqt format using the M.G.L. application to be used in the future. Vina was then launched from a command prompt using the command line. During the configuration process, the grid point spacing was set to its default value of 0.431 angstroms, and the exhaustiveness was set to 8. PyMol and the Discovery Studio Visualizer 2021 were used to examine the output files, which were saved in the pdbqt format. The ligand binding was tested for validity and improved upon using the co-crystallized ligand [25].

   The specific molecular mechanism of the target protein is responsible for the binding of Lupenone. This work aims to determine the inhibitory concentration of each candidate molecule and to find the molecule that, according to the results of the virtual screening, is the most effective at interacting with 2AYD. Using the steepest descent method (1000 steps), which was followed by the addition of the AMBER ff4 force field, the structure of 2AYD was simplified so that it could be more easily understood. This needed to be taken care of before the docking study with the essential ligands could get underway. Before beginning, the investigations into the interaction, the protonation states of the 2AYD that would be involved were tested for neutralization. This was done before beginning the investigations. With the assistance of Auto-Dock version 4.2.6 [27, 28], researchers could conduct experiments on molecular Docking. Polar hydrogen bonds, Kollman and Gastieger charges, and other electrostatic forces were combined to produce the receptor and ligands. After merging the nonpolar hydrogens, the receptor and ligand molecules were finally saved in the pdbqt format. A grid box was created with the values X = 12, Y = 20, and Z = 30, with a spacing of 0.54 angstrom. Lamarckian Genetic Algorithm was used to dock protein-ligand complexes to get the lowest binding free energy (ΔG).

## Molecular dynamic simulations

Schrodinger, L.L.C.'s Desmond 2020.1 was used to carry out 100 ns M.D. simulations of the main protein, 2AYD, in conjunction with the ligand Lupenone (**C.I.D: 643732**). The explicit solvent model with S.P.C. water molecules and the OPLS-2005 force field were utilized [26–28]. In order to get rid of the charge, several Na+ ions were administered. It was decided to add 0.15M NaCl solutions to the system to simulate the physiological environment. The N.P.T. ensemble was generated in each simulation by applying the Nose-Hoover chain coupling method [29, 30]. The simulations were run with the following parameters: a temperature of 300 K; a relaxation period of 1.0 ps; a pressure of 1 bar, and a time step of 2 ps after that. The Martyna–Tuckerman–Klein chain coupling system barostat approach was utilize [31], and the relaxation duration was set at 2 ps. This technique was used to control the pressure. To predict long-range electrostatic interactions, the Colombian interaction radius was fixed at 9, and the particle mesh Ewald technique was utilized [32]. The RESPA integrator was used to ascertain the forces that were not bonded. The root means square deviation was used to evaluate the M.D. simulations' capacity to maintain stability (RMSD).

## Molecular Mechanics Generalized Born and Surface Area (MMGBSA) calculations

During MD simulations of 2ALA complexed with Lupenone, the binding free energy (Gbind) of docked complexes was calculated using the premier molecular mechanics generalized Born surface area (MM-GBSA) module (Schrodinger suite, LLC, New York, NY, 2017–4). The binding free energy was calculated using the OPLS 2005 force field, VSGB solvent model, and rotamer search methods [33]. After the MD run, 10 ns intervals were used to choose the MD trajectories frames. The total free energy binding was calculated using Eq 1:

$$\Delta Gbind = Gcomplex - (Gprotein + Gligand) \qquad (1)$$

Where, ΔGbind = binding free energy, Gcomplex = free energy of the complex, Gprotein = free energy of the target protein, and Gligand = free energy of the ligand. The MMGBSA outcome trajectories were analyzed further for post dynamics structure modifications.

## Results & investigations

### Virtual screening of compounds

The binding affinity of the complex has a score of -9.64 kcal/mol, and it is denoted as 2AYD-643732. The most promising molecule underwent additional reassembly in the binding cavity of 2AYD (Fig 2). The receptor protein 2AYD, shown in **Table 1** with an RMSD score of 0.456 angstroms, was screened using ten different ligands. This computational analysis of binding energy provided us with a clear picture of the ligand that has the best possible affinity with the protein that was being investigated.

### Molecular docking investigation

Molecular Docking is a method that can be employed to ascertain the intermolecular framework that is optimally shaped by a macromolecule in conjunction with medication or added small molecular contender [34]. At the outset, molecular docking research was conducted to screen for and locate the intermolecular interaction that would be most beneficial between the protein of interest and the phytochemical substances. PyRx instruments: To carry out molecular Docking between 10 phytochemical compounds with a three-dimensional structure and

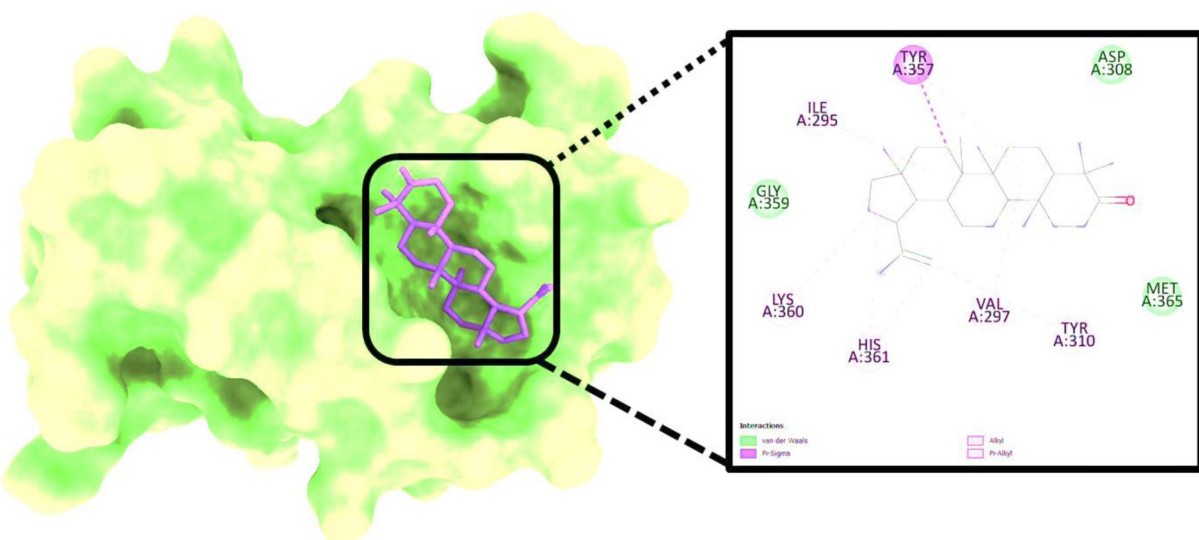

**Fig 2. Analysis of the docked posture of 2AYD-643732; displayed the ligand bound at the pocket of the receptor 2AYD, and the binding pocket residues interacted with the ligand displayed.**

desired protein, the AutoDock Vina wizard has been used. **Table 1** shows the binding affinities of 10 different phytochemicals. Molecular interaction in re-docking studies of ligand Lupenone with 2AYD displayed well-defined binding pocket constituted of residues leucine, valine, proline, alanine, asparagine, serine, cystine, leucine, glycine where the ligand bound to the core of the pocket with binding energy ($\Delta G$) -9.64 kcal/mol and inhibitory concentration ($K_i$) 0.12 mM.

## Molecular dynamic simulation study

The ligand Lupenone (**PubChem ID: 643732**) and the 2AYD protein were subjected to M.D. simulation studies for one hundred nanoseconds to examine the overall quality and stability of the complex till convergence. The root means square deviation (RMSD) of the C-α, the backbone of 2AYD with ligand coupled complex, revealed an extremely stable structure, with a fluctuation of only 1.89 Å. On the other hand, the RMSD of the ligand Lupenone was initially slightly distorted. However, it remained stable until 100 ns without significant variations

**Table 1. Molecular docking of 10 selected phyto-compounds.**

| Sl No | Compound ID (C.I.D.) | Binding energy (Kcal/mol) |
|---|---|---|
| 1. | CID_100332 | -5.6 |
| 2. | CID_101389368 | -5.5 |
| 3. | CID_119034 | -6.4 |
| 4. | CID_15559069 | -7.1 |
| 5. | CID_241572 | -7.0 |
| **6.** | **CID_92158** | **-9.64** |
| 7. | CID_3981577 | -6.9 |
| 8. | CID_5280343 | -5.3 |
| 9. | CID_5280443 | -7.0 |
| 10. | CID_5280445 | -6.2 |

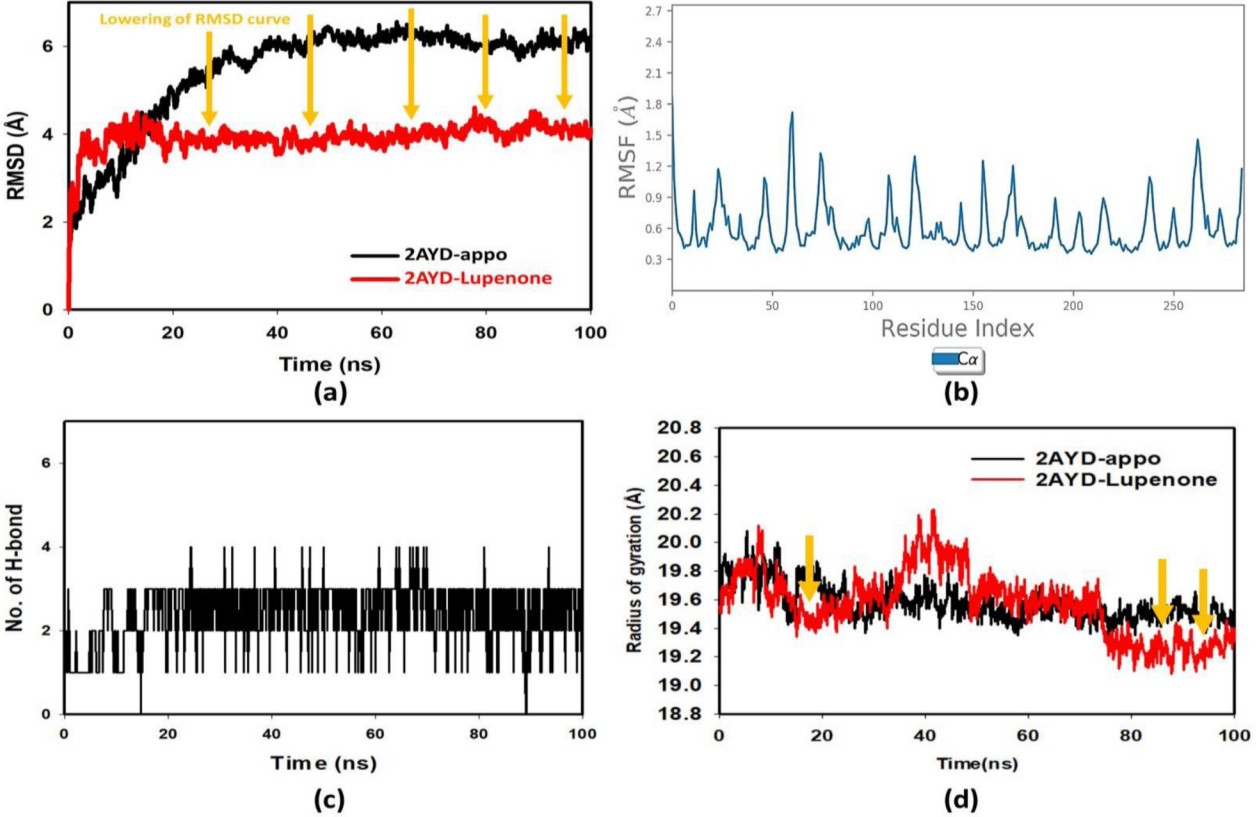

**Fig 3.** **(a)** RMSD of 2AYD and ligand Lupenone for 100 ns; **(b)** RMSF of 2AYD and ligand Lupenone for 100 ns; (c) Number of Hydrogen bindings of 2AYD and ligand Lupenone for 100 ns; (d) Radius of gyration of 2AYD and ligand Lupenone for 100 ns.

(depicted in **Fig 3A**). On the other hand, the root means square fluctuations (RMSF) of the different amino acids that make up the C-α backbone of 2AYD showed the most negligible fluctuations, which indicates that the protein structure is stable. The relative means squared deviation of Lupenone-bound protein simulation trajectories across a timescale of 100 ns. Every piece of data is measured three times to ensure accuracy, and the Y-axis is shifted after each iteration. After a total of 100 ns, the final structure of 2AYD had significant deviations from the reference structure, with an average difference of 2 between it and residue positions 230–250. (Depicted in **Fig 3B**). **Fig 3C** visually represents the typical number of hydrogen bonds established between Lupenone and the various proteins throughout the 100 ns simulation. The MD simulation of Lupenone and 2AYD revealed a significant number of hydrogen bonds. Throughout the simulation, a total of two hydrogen bonds were established. The increased number of hydrogen bonds between protein 2AYD and Lupenone has helped to strengthen the binding and enhance the drought resistance, which has contributed to the simulation's success in maintaining its stability. In addition, the radius of gyration, also known as Rg, was calculated. Rg is an indicator of the size and compactness of the protein when it is in a state where it is attached to the ligand. **Fig 3(D)** depicts the Rg plots for convenience. According to the Rg plot of the C-α backbone, the 2AYD protein displays Rg values ranging from 27.8 to 28.0 angstroms, suggesting significant compactness with an average of 0.3 angstroms from the start to the finish of the 100 ns simulation.

Ligand interaction of Lupenone with predicted docked residues of 2AYD demonstrated the establishment of substantial hydrogen bonds. Apart from this, other non-bonded interactions,

## Protein-Ligand Contacts

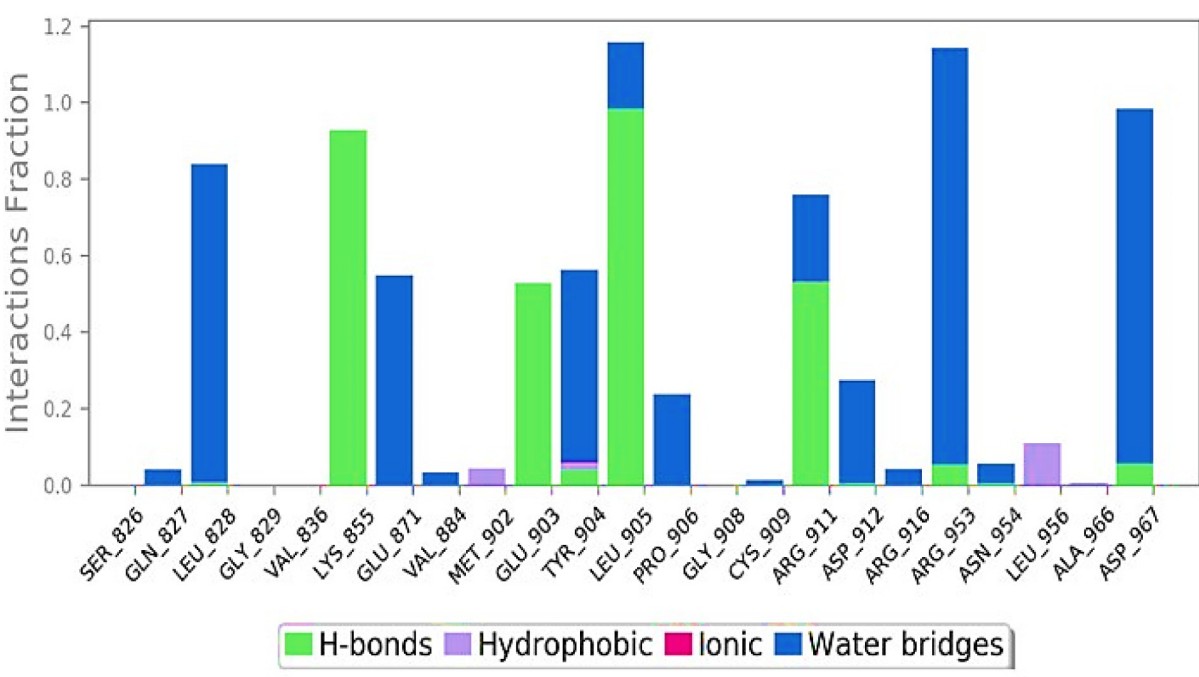

**Fig 4. Types of bonds formed in 100ns simulation run.**

such as hydrophobic interaction and water bridges (illustrated in **Fig 4**). These interactions interplayed a critical role in making a stable complex between the protein and the ligand.

The illustration in **Fig 5A** highlights ligand characteristics such as RMSD, the radius of gyration (rGyr), intramolecular hydrogen bond, molecular surface area (MolSA), solvent accessible surface area (SASA), and polar surface area (P.S.A.). In the ligand, no intramolecular

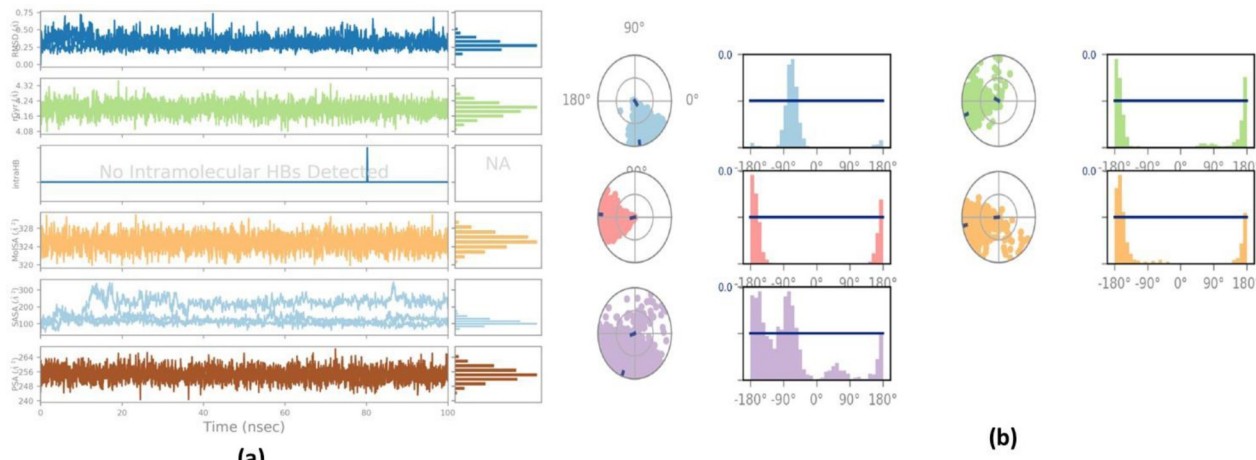

**Fig 5. (a)** The figure displayed shows ligand characteristics such as RMSD, the radius of gyration (rGyr), intramolecular hydrogen bond, molecular surface area (MolSA), solvent accessible surface area (SASA), and polar surface area (P.S.A.) of Lupenone; **(b)** Ligand torsion profile after 100 ns simulation.

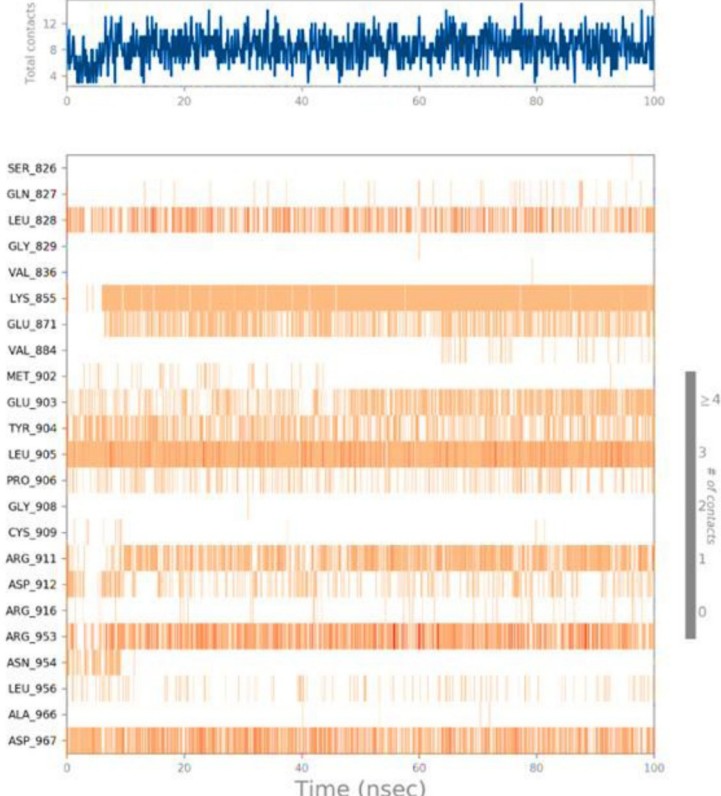

**Fig 6. Protein interactions with the ligand can be monitored throughout the simulation.**

hydrogen was found. In **Fig 5B**, ligand torsion map shows the structural evolution of each rotatable bond (R.B.) over time (0.00 through 100.00 ns). Top: Two-dimensional graphic illustrating rotatable ligand linkages. Dial plots and bar plots, both the same color, indicate rotatable bonds. As the simulation progresses, dial (or radial) charts depict the torsion's conformation. The simulation is traced in a circle from the display's center. Dial plots and bar charts depict the torsion's probability distribution. An infographic shows rotatable bond strength (if potential torsional information is given). Check the chart's left Y-axis for available values. When conducting this research, monitoring the histogram, torsion potential, and protein's conformation strain is crucial to evaluate if the bound shape is maintained. Protein-ligand interaction for different amino acid residues are crucial to docking has been depicted in Fig 6.

## Molecular Mechanics Generalized Born and Surface Area (MMGBSA) calculations

To assess the binding energy of ligands to protein molecules, the MMGBSA technique is commonly employed. The binding free energy of each protein-Lupenone complex, as well as the impact of other non-bonded interactions energies, were estimated (Table 2). With 2AYD, the ligand Lupenone has a binding energy of -50.268 kcal/mol. non-bonded interactions like GbindCoulomb, Gbind Covalent, Gbind Hbond, Gbind Lipo, GbindSolvGB, and GbindvdW govern Gbind. Across all types of interactions, the GbindvdW, GbindLipo, and GbindCoulomb energies contributed the most to the average binding energy. On the other side, the

**Table 2. Binding energy calculation of Lupenone with 2AYD and non-bonded interaction energies from MMGBSA trajectories.**

| Energies (kcal/mol) * | 2AYD |
|---|---|
| $\Delta G_{bind}$ | -550.268 |
| $\Delta G_{bind}Lipo$ | -14.1400 |
| $\Delta G_{bind}vdW$ | -46.4764 |
| $\Delta G_{bind}Coulomb$ | -22.15048 |
| $\Delta G_{bind}H_{bond}$ | -9.686651 |
| $\Delta G_{bind}SolvGB$ | 29.52873 |
| $\Delta G_{bind}Covalent$ | 9.224784 |

GbindSolvGB and Gbind Covalent energies contributed the least to the final average binding energies. Furthermore, the GbindHbond interaction values of Lupenone-protein complexes demonstrated stable hydrogen bonds with amino acid residues. In all of the compounds, Gbind SolvGB and Gbind Covalent exhibited unfavorable energy contributions, and so opposed binding. These conformational changes lead to better binding pocket acquisition and interaction with residues, which leads to enhanced stability and binding energy.

Thus MM-GBSA calculations resulted, from MD simulation trajectories well justified with the binding energy obtained from docking results moreover, the last frame (100 ns) of MMGBSA displayed the positional change of the Lupenone as compared to 0 ns trajectory signify the better binding pose for best fitting in the binding cavity of the protein.

The free energy landscape of (FEL) of achieving global minima of Cα backbone atoms of proteins with respect to RMSD and radius of gyration (Rg) are displayed in **Fig 7**. 2AYD bound to Lupenone achieved the global minima (lowest free energy state) at 2.77 Å and Rg 32 Å (**Fig 7**). The FEL envisaged for deterministic behaviour of 2AYD to lowest energy state owing to its high stability and best conformation at Lupenone bound state. Therefore, FEL is

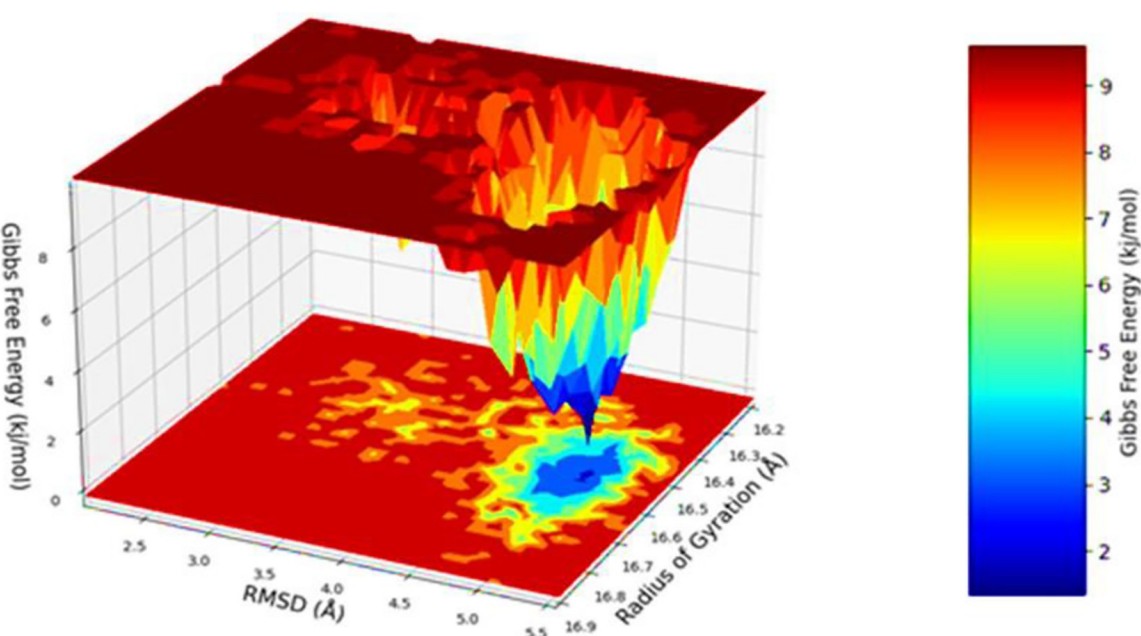

**Fig 7. Free energy landscape displaying the achievement of global minima (ΔG, kj/mol) of 2AYD in presence of Lupenone with respect to their RMSD (nm) and radius of gyration (Rg, nm).**

the indicator of the protein folding to attain minimum energy state, and that aptly achieved due to Wedelosin bound state.

## Discussion & conclusion

Depending on the severity and duration of the stress, abiotic factors (drought, flood, heat, and salt) significantly impact tomato production, causing yield losses of up to 70%. Drought is the most harmful abiotic stress, and tomatoes are susceptible since they lack special genes for drought stress tolerance. Just 20% of agricultural land is irrigated globally, and only 14.5% of it is adequately watered. Under these circumstances, droughts frequently occur, which severely restricts the yield predicted from genetic potential and breeding value [35]. The two main ways that drought alters the relationship between plants and water are the formation of R.O.S. and tomato plants' physiology [36, 37]. Many cultivated and wild species have the gene(s) for drought tolerance. Still, it is difficult to utilize them due to the wide genetic gap between them and the barriers inhibiting embryonic development both before and after transcription. However, computational aid was used to alter the host tomato transcriptional gene(s) for drought stress tolerance. The severity of droughts is the one abiotic element that has a significant impact on agricultural output. Since they are immobile and anchored in one location, plants are vulnerable to environmental and human effects. It is essential to understand how plants react to dry environments to create more drought-resistant cultivars. A large range of drought-responsive proteins have been identified through proteomic analysis of leaf and root tissues; each of these proteins is regulated by a particular set of genes in the plant's genome. In response to biotic and abiotic stimuli, the WRKY gene family members, one of the most diverse sets of plant transcription factors, are essential for plant growth and development. More significantly, WRKY genes regulate how plants respond to pressures brought on by humans and the environment. For the development of plants with greater resistance to biotic and abiotic stressors, it is essential to understand the complex interactions between WRKY proteins and other cellular components, such as other proteins and ligands. The level to which we can accomplish this depends on our knowledge.

The authors of this study suggest controlling transcription factor activity in response to Lupenone, a chemical compound, to boost plant resistance to drought and other stresses. We performed molecular dynamics simulations to examine the predicted structure's unusual behavior when immersed in water. As an added bonus, the structure can be used to gauge the level of difficulty of any ligand or molecule intended to interact with the WRKY transcription protein. Lupenone has a strong affinity for the WRKY protein, with a binding energy of -9.64 kcal/mol. Molecular Docking and modeling were utilized to determine that polyphenols are a key factor in improving drought tolerance in *Solanum Lycopersicum*. In order to characterize the drug's binding affinity to the drought transcription factor-associated target, knowledge of the binding energies of the ligand-target interaction is essential. At long last, we've identified a Phyto-agent, Lupenone, that may increase drought tolerance in cultivated tomato plants.

## Acknowledgments

All the authors acknowledge their respective departments for carrying out this research. S.A. would like to acknowledge Department of Pharmacy, Faculty of Allied Health Sciences, Daffodil International University, Ashulia, Dhaka, Bangladesh (Research4Life Group A Country). Also, authors are grateful to the Deanship of Scientific Research at King Khalid University for funding this study through the Large Research Group Project, under grant number RGP 2/ 100/43.

## Author Contributions

**Conceptualization:** Sandip Debnath, Shopnil Akash.

**Data curation:** Sandip Debnath, Shopnil Akash.

**Formal analysis:** Sandip Debnath, Shopnil Akash.

**Methodology:** Sandip Debnath, Shopnil Akash.

**Project administration:** Sandip Debnath, Hanan M. Alharbi.

**Supervision:** Sandip Debnath, Taha Alqahtani, Ali Alqahtani.

**Validation:** Sandip Debnath, Taha Alqahtani.

**Visualization:** Sandip Debnath.

**Writing – original draft:** Sandip Debnath, Taha Alqahtani, Shopnil Akash.

**Writing – review & editing:** Ali Alqahtani, Hanan M. Alharbi.

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
