## [Decision Letter · Decision Letter 0]

10 Jan 2023

PONE-D-22-34473Lupenone, a wonder chemical obtained from Euphorbia segetalis to boost affinity for the transcriptional factor escalating drought-tolerance in Solanum lycopersicum: a cutting-edge computational biology approachPLOS ONE

Dear Dr. Akash,

Thank you for submitting your manuscript to PLOS ONE. After careful consideration, we feel that it has merit but does not fully meet PLOS ONE’s publication criteria as it currently stands. Therefore, we invite you to submit a revised version of the manuscript that addresses the points raised during the review process.

We look forward to receiving your revised manuscript.

Kind regards,

Arabinda Ghosh

Academic Editor

PLOS ONE

Journal Requirements:

Reviewers' comments:

Reviewer's Responses to Questions

**Comments to the Author**

1. Is the manuscript technically sound, and do the data support the conclusions?

Reviewer #1: Yes

Reviewer #2: Yes

2. Has the statistical analysis been performed appropriately and rigorously? 

Reviewer #1: Yes

Reviewer #2: Yes

3. Have the authors made all data underlying the findings in their manuscript fully available?

Reviewer #1: Yes

Reviewer #2: Yes

4. Is the manuscript presented in an intelligible fashion and written in standard English?

Reviewer #1: Yes

Reviewer #2: Yes

5. Review Comments to the Author

Reviewer #1: The manuscript is very novel and sounds very interesting, few comments must be answered for its better enhancement:

1. Authors must take care of typo errors and grammar errors.

2. Authors should add MMGBSA in their study.

3. Authors must add few more details of drought and its impact.

Reviewer #2: The manuscript seems interesting and paves a new approach to plant drought tolerance, but needs some modifications as follows:

1. The abstract must be concise;

2. A graphical abstract must be included to potray the flow of work;

3. Is there any possibility of commercializing Lupenone???;

4. Make the introduction more crisp which may reflect the cause and the new invention in a flowchart manner;

5. Increase the DPI of RMSDs;

6. Add the data of the Free energy landscape;

7. Add binding free energy calculation by MMGBSA/MMPBSA.

6. PLOS authors have the option to publish the peer review history of their article (what does this mean?). If published, this will include your full peer review and any attached files.

Reviewer #1: **Yes: **Swastika Maitra

Reviewer #2: No

---

## [Author Response · Author response to Decision Letter 0]

16 Jan 2023

Authors Responses to Reviewer

Comments to the Author

Reviewer #1: The manuscript is very novel and sounds very interesting, few comments must be answered for its better enhancement:

1. Authors must take care of typo errors and grammar errors.

Response: Thank you for the progressive comment, we have corrected all typos and grammar errors.

2. Authors should add MMGBSA in their study.

Response: Thank you for the progressive comment, we have added MMGBSA in our revised version.

3. Authors must add few more details of drought and its impact.

Response: Thank you for the progressive comment, we have added some more information about it.

Reviewer #2: The manuscript seems interesting and paves a new approach to plant drought tolerance, but needs some modifications as follows:

1. The abstract must be concise;

Response: Thank you for the progressive comment, we have made it crisp.

3. Is there any possibility of commercializing Lupenone???

Response: Yes, it can easily commercialize.

4. Make the introduction more crisp which may reflect the cause and the new invention in a flowchart manner;

Response: Thank you for the progressive comment, we have made it crisp.

5. Increase the DPI of RMSDs;

Response: We have increased the DPI’s

6. Add the data of the Free energy landscape;

Response: Thank you for the progressive comment, we have added it in revised version.

7. Add binding free energy calculation by MMGBSA/MMPBSA

Response: Thank you for the progressive comment, we have added MMGBSA in our revised version.

---

## [Decision Letter · Decision Letter 1]

20 Jan 2023

Lupenone, a wonder chemical obtained from Euphorbia segetalis to boost affinity for the transcriptional factor escalating drought-tolerance in Solanum lycopersicum: a cutting-edge computational biology approach

PONE-D-22-34473R1

Dear Dr. Akash,

We’re pleased to inform you that your manuscript has been judged scientifically suitable for publication and will be formally accepted for publication once it meets all outstanding technical requirements.

Kind regards,

Arabinda Ghosh

Academic Editor

PLOS ONE

Reviewers' comments:

Reviewer's Responses to Questions

**Comments to the Author**

1. If the authors have adequately addressed your comments raised in a previous round of review and you feel that this manuscript is now acceptable for publication, you may indicate that here to bypass the “Comments to the Author” section, enter your conflict of interest statement in the “Confidential to Editor” section, and submit your "Accept" recommendation.

Reviewer #1: All comments have been addressed

Reviewer #2: All comments have been addressed

2. Is the manuscript technically sound, and do the data support the conclusions?

Reviewer #1: Yes

Reviewer #2: Yes

3. Has the statistical analysis been performed appropriately and rigorously? 

Reviewer #1: Yes

Reviewer #2: Yes

4. Have the authors made all data underlying the findings in their manuscript fully available?

Reviewer #1: Yes

Reviewer #2: Yes

5. Is the manuscript presented in an intelligible fashion and written in standard English?

Reviewer #1: (No Response)

Reviewer #2: Yes

6. Review Comments to the Author

Reviewer #1: (No Response)

Reviewer #2: The authors have addressed all the comments very satisfactory.

The article is suitable for publication.

7. PLOS authors have the option to publish the peer review history of their article (what does this mean?). If published, this will include your full peer review and any attached files.

Reviewer #1: **Yes: **Swastika Maitra

Reviewer #2: No

---

## [Editor Report · Acceptance letter]

20 Apr 2023

PONE-D-22-34473R1 

Lupenone, a wonder chemical obtained from *Euphorbia segetalis* to boost affinity for the transcriptional factor escalating drought-tolerance in *Solanum Lycopersicum*: a cutting-edge computational biology approach 

Dear Dr. Akash:

I'm pleased to inform you that your manuscript has been deemed suitable for publication in PLOS ONE. Congratulations! Your manuscript is now with our production department. 

Kind regards, 

on behalf of

Dr. Arabinda Ghosh 

Academic Editor

PLOS ONE